# Alcohol and HIV-Derived Hepatocyte Apoptotic Bodies Induce Hepatic Stellate Cell Activation

**DOI:** 10.3390/biology11071059

**Published:** 2022-07-14

**Authors:** Moses New-Aaron, Raghubendra Singh Dagur, Siva Sankar Koganti, Murali Ganesan, Weimin Wang, Edward Makarov, Mojisola Ogunnaike, Kusum K. Kharbanda, Larisa Y. Poluektova, Natalia A. Osna

**Affiliations:** 1Department of Environmental Health, Occupational Health and Toxicology, College of Public Health, University of Nebraska Medical Center, Omaha, NE 68198, USA; 2Research Service, Veterans Affairs Nebraska-Western Iowa Health Care System, Omaha, NE 68105, USA; raghu.dagur82@gmail.com (R.S.D.); skoganti@unmc.edu (S.S.K.); murali.ganesan@unmc.edu (M.G.); mogunnaike@unmc.edu (M.O.); kkharbanda@unmc.edu (K.K.K.); 3Department of Internal Medicine, University of Nebraska Medical Center, Omaha, NE 68105, USA; 4Department of Pharmacology and Experimental Neuroscience, University of Nebraska Medical Center, Omaha, NE 68105, USA; weimin.wang@unmc.edu (W.W.); makarove@unmc.edu (E.M.); lpoluekt@unmc.edu (L.Y.P.)

**Keywords:** hepatocytes, apoptotic bodies, hepatic stellate cells, JNK-ERK1/2, reactive oxygen species, JAK-STAT3, acetaldehyde-generating system, ethanol

## Abstract

**Simple Summary:**

Did you know that alcohol abuse among people living with HIV (PLWH) is twice as frequent as in the general population? This undoubtedly makes alcohol abuse in the context of HIV a severe public health issue, which is often attributed to several comorbidities including liver disease. In fact, 50% of mortality from HIV-related liver disease in the United States is attributed to alcohol. Previously, we showed that an alcohol metabolite, acetaldehyde, drives HIV accumulation in hepatocytes, which generates oxidative stress leading to hepatocyte death. When large vesicles (apoptotic bodies) from dead hepatocytes are internalized by hepatic stellate cells (HSC), it causes their profibrotic activation. In fact, our data revealed that upon internalization of HIV-containing hepatocyte apoptotic bodies, HSC generate reactive oxygen species (ROS), which induce fibrosis by activating JNK-ERK1/2 and JAK-STAT3 pathways. Therefore, a potential therapeutic regimen for reducing liver fibrosis among PLWH should include ROS-scavenging antioxidants.

**Abstract:**

Recently, we found that both HIV and acetaldehyde, an alcohol metabolite, induce hepatocyte apoptosis, resulting in the release of large extracellular vesicles called apoptotic bodies (ABs). The engulfment of these hepatocyte ABs by hepatic stellate cells (HSC) leads to their profibrotic activation. This study aims to establish the mechanisms of HSC activation after engulfment of ABs from acetaldehyde and HIV-exposed hepatocytes (AB_AGS+HIV_). In vitro experiments were performed on Huh7.5-CYP (RLW) cells to generate hepatocyte ABs and LX2 cells were used as HSC. To generate ABs, RLW cells were pretreated for 24 h with acetaldehyde, then exposed overnight to HIV1_ADA_ and to acetaldehyde for 96 h. Thereafter, ABs were isolated from cell suspension by a differential centrifugation method and incubated with LX2 cells (3:1 ratio) for profibrotic genes and protein analyses. We found that HSC internalized ABs via the tyrosine kinase receptor, Axl. While the HIV gag RNA/HIV proteins accumulated in ABs elicited no productive infection in LX2 and immune cells, they triggered ROS and IL6 generation, which, in turn, activated profibrotic genes via the JNK-ERK1/2 and JAK-STAT3 pathways. Similarly, ongoing profibrotic activation was observed in immunodeficient NSG mice fed ethanol and injected with HIV-derived RLW ABs. We conclude that HSC activation by hepatocyte AB_AGS+HIV_ engulfment is mediated by ROS-dependent JNK-ERK1/2 and IL6 triggering of JAK-STAT3 pathways. This can partially explain the mechanisms of liver fibrosis development frequently observed among alcohol abusing PLWH.

## 1. Introduction

Human Immunodeficiency Virus (HIV) remains a major global health challenge of epic proportions, with approximately 38 million people living globally with HIV [1], about three percent of which are in the United States [2]. Despite the success of antiretroviral therapy (ART) in combating HIV [3,4,5], alcoholic liver disease remains a paramount cause of morbidity and mortality among people living with HIV (PLWH) [6,7,8]. 

This may be explained by the increased prevalence of alcohol consumption among PLWH [9,10,11]. In fact, alcohol consumption among PLWH is twice as frequent as in the general population [12]. While alcohol misuse is strictly of behavioral etiology, behavioral interventions may not provide an efficient, holistic intervention [9]. Hence, there is a strong need to explore the pathomechanisms of alcohol- and HIV-induced hepatotoxicity. Elucidation of these mechanisms will help with developing treatment modalities for preventing alcohol-induced hepatotoxicity and further progression to end-stage liver disease among PLWH. 

Recently, we demonstrated a “double-edged sword” phenomenon where acetaldehyde, an ethanol metabolite, favorably induced massive apoptosis in HIV-expressing hepatocytes. While clearance of hepatocytes bearing HIV RNA and HIV proteins prevents the accumulation of HIV DNA and its integration in the hepatocyte genome, the generated hepatocyte apoptotic bodies (ABs) adversely activate profibrotic genes when engulfed by hepatic stellate cells (HSC) [13]. More recently, we uncovered oxidative stress-mediated lysosome impairment as the basis for acetaldehyde-induced apoptosis in HIV-expressing hepatocytes [14]. However, the mechanisms of ABs engulfment and profibrotic activation of HSC in the context of HIV and alcohol are not yet understood. 

Hepatic fibrosis is characterized by two dynamic and contrasting events, i.e., excessive extracellular matrix (ECM) deposition accompanied by its decreased degradation. Activation of hepatic perisinusoidal resident non-parenchymal cells, HSC, is a principal event in liver fibrotic development and progression. Quiescent HSC typically contain retinoids. Upon activation, HSC undergo a morphological and functional transformation into α-smooth muscle actin (SMA)-expressing myofibroblasts. Several triggers, including bacterial lipopolysaccharide [15], reactive oxygen species [16], and viruses [17], are known to activate HSC. In addition, multiple studies have identified CpG motifs in DNA as a potent stimulant of HSC through the toll-like receptor (TLR)-9, localized in the endosomal compartment of HSC [18,19,20]. Since hepatocyte apoptosis often accompanies liver fibrosis, it can be attributed to the direct effects of Damage Associated Molecular Patterns, DAMPs, from hepatocyte Abs [21]. 

Although hepatocyte ABs are known to canonically activate HSC via TLR9, ABs from acetaldehyde and HIV exposed hepatocytes (AB_AGS+HIV_), in our hands, surprisingly, did not activate HSC through TLR9. This prompted us to look for other mechanistic explanations for AB_AGS+HIV_-induced HSC activation. Therefore, we first characterized ABs exposed to acetaldehyde and HIV to understand their content. Furthermore, HSCs were characterized to identify the candidate receptors involved in ABs engulfment. Given that AB_AGS+HIV_ is a product of HIV particle accumulation and the subsequent oxidative stress, mitogen activated protein kinases (MAPK) [22,23,24,25], and the JAK-STAT3 pathway [26,27,28], which notoriously respond to signals from reactive oxygen species (ROS), endothelial cell products and viral particles were explored as potential mechanisms for AB_AGS+HIV_-induced HSC activation.

## 2. Materials and Methods

### 2.1. Reagents and Antibodies

Reagents were purchased from the following commercial suppliers: high-glucose Dulbecco’s Modified Eagle Medium (DMEM) and fetal bovine serum (FBS) were obtained from Invitrogen (Waltham, MA, USA); Trizol was purchased from Life Technologies (Carlsbad, CA, USA); primer probes, high-capacity reverse transcription kit, and real-time polymerase chain reaction (RT-PCR) reagents were from Applied Biosystems by Thermo Fisher Scientific, Carlsbad, CA, USA. Pharmacological Inhibitors: TAM Family protein inhibitor, LDC1267, was obtained from MedChemExpress (Monmouth Junction, NJ, USA, HY-12494), JNK inhibitor, SP600125, was obtained from Selleckchem (Houston, TX, USA, S1460), SB202190 for MAPK p38 was obtained from Selleckchem (Houston, TX, USA, S1077). Primary antibodies used were: (a) mouse monoclonal: Anti-HIV-1 p24 Antibody: sc-69728, Anti-Stat3 Antibody (F-2): sc-8019, Anti-β-Actin Antibody (C4): sc-47778 (Santa Cruz Biotechnology, Dallas, TX, USA), anti-smooth muscle actin: MAB1420-SP, anti-human proteinS: AMB4036-SP (R&D Systems, Minneapolis, MN, USA). Anti-Nef antibody: ARP-3689, Anti-HIV1 Tat antibody: ARP7377 (American Type Culture Collection, Manassas, VA, USA) (b) rabbit monoclonal and polyclonal: anti-HIV-1 Vpr (American Type Culture Collection, Manassas, VA, USA, ARP-11836); anti-Stat1 antibody (Santa Cruz Biotechnology, Dallas, TX, USA, Sc-592), anti-phospho-ERK1/2: 9101, anti-ERK1/2: 4694, phosphor-STAT1:9177 (Cell Signaling Technology, Danvers, MA, USA), Anti-Cytochrome P450 Enzyme CYP2E1 Antibody: AB1252 (Millipore Sigma, Burlington, MA, USA) (c) Goat polyclonal: anti human AXL antibody: AF154-SP, anti-human Gas6 antibody: AF885-SP (R&D Systems, Minneapolis, MN, USA) (d) secondary antibodies: (I) IRDye 680RD Goat anti-Rabbit, C50317-02; IRDye 680RD Donkey anti-Mouse, C50520-02; (II) Goat anti-Mouse IgG (H + L) Cross-Adsorbed Secondary Antibody, Alexa Fluor 555, Carlsbad, CA, USA, A21422; and Donkey anti-Rabbit IgG (H + L) Highly Cross-Adsorbed Secondary Antibody, Alexa Fluor 488.

### 2.2. In Vitro Studies

Since our experiments required cell-to-cell communication, we performed most experiments on two major human cell lines: hepatocyte-like RLW cells (Huh7.5 cells stably transfected with CYP2E1) and HSC-like LX2 cells. We used these cell lines because primary human hepatocytes (PHH) and HSC are not readily available. RLW cells were characterized by attenuated innate immunity, making them suitable for demonstrating the HIV-infection effects. However, RLW cells do not express alcohol dehydrogenase (ADH) and only metabolize ethanol by CYP2E1, generating very low amount of acetaldehyde. Therefore, to mimic ethanol metabolism in RLW cells as observed in primary human hepatocytes, we developed an exogenous, continuous source of acetaldehyde because the observed pro-apoptotic effect of ethanol comes from ethanol metabolism. The exogenous supply of acetaldehyde, called the acetaldehyde-generating system (AGS), was introduced to the cell culture medium. AGS continuously produces physiological amounts of acetaldehyde when 0.02 EU yeast ADH metabolizes 50 mM ethanol in the presence of 22 mM nicotinamide adenine dinucleotide (NAD) as a co-factor [29]. 

As a source of HIV, primary human macrophage-propagated HIV-1_ADA_ was purified at University of Nebraska Medical Center (UNMC) [14]. The LX2 cell line, derived from human HSC, was generously provided by Dr. Laura Schrum (Carolinas Healthcare System, Charlotte, NC, USA), who originally obtained the cell line from Dr. Scott Friedman (Icahn School of Medicine at Mount Sinai, NY, USA).

To generate the required ABs, RLW cells were pretreated or not for 24 h with AGS, infected or not with 0.1 MOI HIV 1_ADA_ overnight, and then exposed to AGS or not for 96 h. HIV in culture medium and cell membrane-bound HIV were removed from the culture system by copiously washing the HIV-exposed RLW cells with DMEM at least three times. This 96 h exposure to AGS after the treatment of cells with HIV induces robust apoptosis in RLW cells. 

Isolated RLW ABs were introduced to LX2 cells grown in 2% FBS-containing DMEM. RLW Abs, quantified by nanoparticle tracking analysis (NTA), were introduced to LX2 cells at the ratio of 3:1 and incubated for either 2 h or 48 h depending on the endpoint measurement. mRNA measurements were obtained from LX2 cells exposed to RLW ABs for 2 h, while proteins were measured in LX2 cells exposed to RLW ABs for 48 h. 

### 2.3. Isolation of ABs from RLW Cells

Isolation of ABs from RLW cells was performed by a differential centrifugation technique as previously reported [30]. Briefly, AB-containing HIV-exposed and AGS-pretreated RLW cell supernatants were collected in a 50 mL conical tube and then centrifugated at 300× *g* for 10 min to pellet the cell debris. The cell debris-free supernatants were collected into a new 50 mL conical tube and then centrifuged at 3000× *g* for 20 min to pellet RLW ABs. The pelleted RLW ABs were separated from the exosome and microvesicle-rich supernatant. To avoid exosome or microvesicle contamination, the pelleted ABs were washed three times with FBS-free DMEM. Figure 1 provides the schematic illustration of the involved process. 

### 2.4. Generation of TAMRA-Labeled RLW ABs by Prolonged Incubation

RLW cells were cultured as described in Section 2.2. Before the final 96 h incubation of RLW cells with AGS, 10 µM TAMRA-succinimide ester was added to the RLW media and incubated at 37 °C for 30 min. Then excess TAMRA dye was removed by washing the cells thrice with copious media, and the TAMRA-labeled cells were left to incubate. After 96 h of incubation, TAMRA-labeled AB-containing supernatants were collected from the culture flask or dish and processed as indicated in Section 2.3

### 2.5. siRNA Transfection of LX2 Cells

Inhibition of STAT3 expression in LX2 cells was achieved by STAT3 siRNA (sc-29493, Santa Cruz, Dallas, TX, USA) transfection according to the manufacturer’s instructions. The siRNA transfection efficiency was evaluated using scrambled siRNA-FITC Conjugate-A (sc-36869, Santa Cruz, Dallas, TX, USA) and immunostaining was conducted to detect signal-positive cells. Four hours before the siRNA (STAT3 or scrambled) application, FBS-containing media (DMEM, 2% FBS and 1% Penicillin streptomycin) were replaced with FBS-free media (DMEM and 1% Penicillin streptomycin) in the LX2 culture system. Thereafter, the cells were incubated with siRNA in transfection media (sc-36868, Santa Cruz, Dallas, TX, USA) for 6 h. To evaluate the effects of AB_AGS+HIV_ from RLW cells on LX2 cells after STAT3 genes silencing, AB_AGS+HIV_ were introduced to LX2 cells at the ratio of 3:1 for 2 h or 48 h to measure changes in profibrotic genes and proteins. Control siRNA (sc-37007, Santa Cruz, Dallas, TX, USA) was included in the experiment as a negative control. 

### 2.6. Imaging of ABs by Transmission Electron Microscopy

ABs derived from RLW cells were isolated as described in Section 2.3. To study the size and morphology of the ABs, an isolated sample was submitted to the Electron Microscopy Core Facility at the UNMC. A FEI Technai G2 Spirit transmission electron microscope was used to evaluate AB sizes and morphologies. 

### 2.7. In Vivo Studies

For validation purposes, immunodeficient NSG mice (NOD.Cg-Prkdcscid Il2rgtm1Wjl/SzJ, NSGTM Stock No: 005557, The Jackson Laboratories) were used for evaluating the ability of HIV-infected ABs to promote profibrotic changes when engulfed by LX2 cells. Immunodeficient NSG mice were used because (I) the exclusion of a species-specific immune response elicited between mouse HSC and human hepatocyte-derived ABs is necessary, (II) previous experience working on NSG mice and evidence from already published works showed that NSG mice have a high tolerance for alcohol administration [31], and (III) NSG mice are natural killer (NK) cell deficient, which is beneficial for our model because NK cells in the presence of ABs can interfere with liver fibrosis [32]. 

Immunodeficient NSG male mice, 8–10 weeks old, were fed with a nutritionally adequate liquid diet containing 5% ethanol or a pair fed diet in which ethanol was isocalorically substituted with dextran maltose (BioServ, Frenchtown, NJ, USA). For the ethanol group, ethanol was introduced gradually by increasing ethanol content by 1% (*v/v*) every day until the mice attained a consumption of 5% ethanol (*v/v*). This feeding was continued for ten weeks. To study the role of HIV-containing ABs and alcohol in fibrosis development, mice were subdivided into the following groups:Control-fed mice exposed to (A) uninfected and (B) HIV-infected ABs generated from RLW cells.Ethanol-fed mice exposed to (A) uninfected and (B) HIV-infected ABs from RLW cells. Each subgroup contains 3 mice.

Given that previous fibrosis studies supplemented ethanol diet with carbon tetrachloride, CCl4, injections to attain liver fibrosis in NSG mice [33], 2.5 mL/kg body weight of 10% CCl4 (Sigma, St. Louis, MO, USA) dissolved in olive oil was injected intraperitoneally three times per week during the last two weeks of ethanol/control feeding. This corresponded to the period of RLW ABs (0.5–1 × 10^7^/mouse) intraperitoneal injection (i.p). In the 10th week of the experiment, the mice were sacrificed. Liver damage was measured by ALT/AST in serum. Liver tissue was stained by Sirius Red for fibrosis. Profibrotic genes were measured by real time polymerase chain reaction (RT-PCR). 

### 2.8. RNA Isolation and RT-PCR

RNAs encoding HIV gag, TIMP1, TGFβ1, TGFβ2, COL1A1, MMP2, SOCS1, ACTA2, APOBEC, OAS1, ISG15, and IL6 were measured by RT-PCR as previously described [14]. Total cellular RNA was isolated from cells using Trizol reagent. This involved a two-step procedure in which at least 200 ng of RNA was transcribed to cDNA using a high-capacity reverse transcription kit (Applied Biosystems, ThermoFisher Scientific, Waltham, MA USA) and the cDNA transcript was amplified using TaqMan Universal Master Mix II with fluorescent-labeled primers (TaqMan gene expression systems) using a Model 7500 qRT-PCR thermal cycler. The relative quantity of each RNA transcript was calculated by its threshold cycle (Ct) after subtracting the reference, GAPDH.

### 2.9. Immunoblotting

Immunoblotting was performed as described [13]; however, in this study, blots were developed using 27444 Bio-Rad Imaging System Chemidoc Touch Imaging System and protein band densities were quantified using the National Institute of Health (NIH) Image J software program. Equal amounts of protein (20 µg) were loaded in each lane. β-actin was used as the loading control for normalization. All original blots were presented in the Appendix A.

### 2.10. Immunofluorescence

LX2 cells (15,000/well) were seeded onto a 16-well chamber slide. RLW ABs were introduced to LX2 cells at the ratio of 3:1. After overnight incubation of cells with RLW ABs, the RLW ABs containing medium were removed and replaced with fresh media. LX2 cells were incubated for another 48 h, then were washed with 1× PBS, fixed with 4% paraformaldehyde for 12 min at 37 °C, permeabilized with 0.1% Triton X-100 for 3 min at room temperature, and blocked for 30 min with 1% (g/vol) bovine serum albumin (BSA) in PBS. LX2 cells were stained to study α-SMA and STAT3. First, cells were incubated with primary antibodies for 1 h, then with Alexa-Fluor-labeled secondary antibodies for 30 min. Nuclei were stained with DAPI. The coverslips were transferred to microscope slides for imaging with a Keyence BZ-X810 fluorescence microscope. Immunostainings were quantified for intensity using the Image J V1.8.0 software program, NIH (Bethesda, MD, USA).

### 2.11. Statistical Analyses

Data were analyzed using GraphPad Prism v7.03 software (GraphPad, La Jolla, CA, USA). Data from at least three duplicate, independent experiments were expressed as mean ± SEM. Comparisons among multiple groups were performed by non-parametric one-way ANOVA and Kruskal–Wallis test, accompanied by a post hoc test. For comparisons between two groups, we used the Mann–Whitney U test. A *p*-value of 0.05 or less was considered statistically significant.

## 3. Results

### 3.1. HIV RNA, HIV Proteins and Malondialdehyde Were Expressed by RLW AB_AGS+HIV_

Since extracellular vesicles (EVs) are established carriers of molecular cargoes known to mediate intercellular communications [34,35,36], it became expedient to uncover the contents of RLW ABs. First, the sizes of the isolated RLW ABs were determined to be 800–1700 nm (Figure 2A) via electron microscopy. This corresponds to previously reported sizes of ABs [37]. Then, we observed through NTA that the combined treatment of acetaldehyde and HIV engendered the highest concentration of RLW ABs (Figure 2B). Since previously observed intense hepatocyte apoptosis was due to acetaldehyde-induced HIV RNA and protein accumulation in hepatocytes, the HIV RNA content of RLW ABs was investigated. We found the highest content of HIV gag RNA in RLW ABs generated from cells treated with both acetaldehyde and HIV (Figure 2C). This was accompanied by upregulated HIV protein (p24, Nef, and Tat, but not Vpr) expressions in RLW AB_AGS+HIV_ (Figure 2D–G). Moreover, we measured the oxidative stress marker, malondialdehyde (MDA) and CYP2E1 (Appendix B
Figure A1A,B) in AB since oxidative stress was the major trigger of RLW apoptosis under the combined treatment of acetaldehyde and HIV as indicated previously [13,14]. Here, MDA detected by immunoblotting analysis was upregulated by 62.5% in AB_AGS+HIV_ as compared to AB_control_ (Figure 2H,I). The purities of the isolated RLW ABs were confirmed by probing the ABs lysate for exosome marker CD63. While CD63 was not expressed by RLW ABs, other apoptotic markers, calnexin, calreticulin (Figure 2J) and phosphatidylserine (PtylS) (Figure 2K), were sufficiently expressed. 

### 3.2. Engulfment of RLW ABs by LX2 Cells

During efferocytosis of ABs, PtylS expressed on ABs are recognized by phosphatidylserine recognition receptors (PtylSRR) [38]. These PtylSRR are either TIM family receptors or TAM family receptors [39,40]. While LX2 cells do not express TIM 1 and 3 (Appendix B
Figure A2A,B), they express TAM receptors. The TAM family receptors consist of Axl, Tyro3 and Mer, sub-family members of receptor tyrosine kinases (RTKs) [40]. As previously reported, Mer is predominantly expressed on macrophages of specific organs but not in the liver [41,42]. Therefore, only Axl and Tyro3 were tested as potential receptor candidates for LX2 cell engulfment of RLW ABs. 

While immunofluorescence staining revealed the expression of both Axl and Tyro3 in LX2 cells, Axl was predominantly localized extranuclearly (Figure 3A), while Tyro3 was localized intranuclearly (Figure 3B). This excludes Tyro3 as the potential TAM receptor candidate responsible for the engulfment of RLW ABs by LX2 cells. Furthermore, effective engulfment via TAM receptors requires intracellularly generated ligands, Gas 6 and Protein S, to act as inserts between the PtylS on ABs and TAM receptors on LX2 cells [43]. To this end, the expressions of these ligands were confirmed by the visualization of Gas 6 and Protein S in LX2 cells (Figure 3C,D). Since LX2 cells express Axl and Gas 6/Protein S, the LX2 engulfment of RLW ABs was demonstrated by exposing TAMRA-labeled ABs to LX2 cells (Figure 3E). 

### 3.3. Pharmacological Inhibition of Axl Blocks LX2 Engulfment of RLW ABs and Attenuates Atcivation of Profibrotic Genes

Given that Axl is the potential TAM receptor for LX2 engulfment of RLW ABs, pharmacological inhibition of Axl with 1µM LDC1267 (Axl inhibitor) reduced the number of engulfed TAMRA-labeled RLW ABs by 6-fold (Figure 4A,B). As an additional confirmation, this inhibitor showed attenuated expression of TAMRA-labeled ABs in LX2 cells, as indicated by flow cytometry (Figure 4C). Consequentially, this inhibitor downregulated HIV gag RNA expression (Figure 4D), SOCS 1 genes (mRNA) (Figure 4E), and suppressed profibrotic genes (Figure 4F–I) in LX2 cells exposed to RLW AB_AGS+HIV_.

### 3.4. Engulfment of HIV- and MDA-Containing ABs Induces LX2 Profibrotic Activation

We explored the possible effects of RLW ABs engulfment by LX2 cells. First, we checked if the HIV gag RNA or HIV proteins were delivered to recipient LX2 cells with RLW ABs. It appeared that, in LX2 cells, the expression of HIV gag RNA was 5.5-fold more after exposure to AB_AGS+HIV_ compared with exposure to RLW AB_HIV_ (Figure 5A). To determine if engulfed RLW AB_AGS+HIV_ can elicit infection in recipient LX2 cells, we exposed AB_AGS+HIV_ at the ratio of 3:1 to permissive-macrophage-like cell THP1 (as a positive control). While we observed upregulation of HIV RNA in LX2 cells with an increased THP1 cell: AB_AGS+HIV_ ratio (Figure 5B), we could not detect the release of p24 to the culture media using ELISA techniques (Figure 5C). Furthermore, a two-time point (Day 1 and 3) kinetic experiment in THP1cells incubated with AB_AGS+HIV_ did not significantly change reverse transcriptase (RT), an HIV-replication enzyme, activity (Figure 5D). However, the level of ROS generated in LX2 cells after internalization of RLW AB_AGS+HIV_ was 45% higher than in AB-non-exposed LX2 cells (Figure 5E). Moreover, significant profibrotic activation was observed in LX2 cells exposed to AB_AGS+HIV_ (Figure 5F–J). Interestingly, when we used AB_AGS+HIV_ not generated from RLW cells but from lymphocyte-like Jurkat cells treated in the same way, we observed no profibrotic activation of LX2 cells (Appendix B
Figure A3A–D). Additionally, ABs derived from AGS and HIV-exposed RLW cells contain no profibrotic genes expressed by HSCs upon AB internalization (Appendix B
Figure A4A,B). 

### 3.5. Pharmacological Inhibition of LX2 Cells Exposed to AB_AGS+HIV_ Attenuates Profibrotic Activation via JNK and ERK1/2 Pathway

Since MDA and HIV-containing AB_AGS+HIV_ triggered ROS generation in LX2 cells and correlates positively with profibrotic gene activation, we hypothesize that c-jun N-terminal Kinase (JNK) and extracellular signal-regulated kinase 1 and 2 (ERK1/2), members of the four distinct mammalian MAPK known to respond to ROS [44], are involved in the profibrotic activation of LX2 cells after engulfment of MDA- and HIV-containing RLW AB_AGS+HIV_. To test our hypothesis, LX2 cells exposed to AB_AGS+HIV_ were pretreated with 10µM JNK inhibitor, SP600125. In this case, TGFβ1, ACTA2, and COL1A1 mRNAs were attenuated by approximately 25% when compared to LX2 cells without the inhibitor (Figure 6A–C). Since JNK is upstream to ERK1/2, JNK inhibitor attenuated phospho-ERK1/2, which is required for nuclear translocation of ERK1/2 to the nucleus to control proliferation and growth of LX2 cells. This consequentially led to the attenuation of α-SMA in LX2 cells (Figure 6D–F). Immunostaining analysis corroborated the findings on the attenuation of α-SMA when LX2 cells were treated with JNK inhibitor (Figure 6G,H). To test if other MAPK family members provided similar effects, LX2 cells exposed to AB_AGS+HIV_ were pretreated with 10µM p38 MAPK inhibitor, SB202190. Surprisingly, phosphoERK1/2, ACTA2, TGFβ1 mRNAs and α-SMA were upregulated in LX2 cells exposed to AB_AGS+HIV_ (Appendix B
Figure A5A–E). 

### 3.6. Oxidative Stress from AB_AGS+HIV_ Activates JNK and ERK1/2 Pathway in LX2 Cells

Oxidative stresses are established signals for the activation of JNK and ERK1/2 pathways to activate critical cellular processes, such as differentiation, growth, cell proliferation, and cell survival [44]. Given that oxidative products are a vital cargo of RLW AB_AGS+HIV_, it becomes evident that oxidative stress contributes to the observed profibrotic activation. This was confirmed when pretreatment with 5 mM N-acetyl cysteine (NAC), a known antioxidant, attenuated TGFβ2 mRNA and TIMP1 mRNA by 50% and 20%, respectively, in RLW AB_AGS+HIV_-treated LX2 cells (Figure 7A,B). NAC further attenuated ERK1/2 phosphorylation in a similar fashion as the JNK inhibitor, thereby resulting in the downregulation of α-SMA (Figure 7C–G).

### 3.7. AB_AGS+HIV_ Upregulates IL6 mRNA in LX2 Cells

Since HIV gag RNA contained in AB _AGS+HIV_ is deposited in LX2 cells after engulfment, it is necessary to investigate the JAK-STAT1 (antiviral) pathway which is required for Interferon Stimulating Genes (ISGs) activation. To induce JAK-STAT1 signaling, LX2 cells were treated with 1000 U/mL interferon IFNα for 30 min in the presence or absence of RLW AB_AGS+HIV_. Here, RLW AB_AGS+HIV_ downregulated the STAT1 phosphorylation induced by IFNα as indicated by the immunoblots (Figure 8A,B). Next, to determine the effects of RLW AB_AGS+HIV_ on ISGs, LX2 cells were treated with 400 U/mL IFNα for 6 h in the presence or absence of RLW AB_AGS+HIV_. RLW AB_AGS+HIV_ attenuated APOBEC-3G (Figure 8C), OAS1 (Figure 8D) and ISG15 (Figure 8E) mRNAs. Moreover, RLW AB_AGS+HIV_ induced the activation of IL6 mRNA (Figure 8F), the gene for IL6 cytokine, which is important for JAK-STAT3 activation.

### 3.8. siRNA STAT3 Transfection Inhibits STAT3 Protein Expressions in LX2 Cells

Since AB_AGS+HIV_ induced the upregulation of IL6 mRNA, a STAT3 ligand gene, it became expedient to explore the role of AB_AGS+HIV_ in the activation of the STAT3 pathway in LX2 cells. To examine the input of the STAT3 pathway in the profibrotic activation of LX2 cells exposed to RLW AB_AGS+HIV_, we inhibited STAT3 genes by STAT3 siRNA transfection. The siRNA transfection efficiency was evaluated with scrambled FITC-labeled siRNA, resulting in 90.25% siRNA FITC-positive cells (Figure 9A,B). Control siRNA transfection was used as a negative control. Then, we cultured LX2 cells with STAT3 siRNA and control siRNA as described in the Section 2. An approximately 11-fold reduction of STAT3 puncta was observed in immunostained LX2 cells transfected with STAT3 siRNA compared to those transfected with control siRNA (Figure 9C,D). Additional analysis from immunoblots revealed the attenuation of STAT3 protein expression by 60% in LX2 STAT3 siRNA transfected cells compared to the cells transfected with control siRNA (Figure 9E,F). 

### 3.9. Silencing STAT3 in LX2 Cells Attenuates AB_AGS+HIV_ Induced Profibrotic Activation

Since AB_AGS+HIV_ induced upregulation of IL6 mRNA after 6 h of incubation with LX2 cells, STAT3 may be involved in the observed AB_AGS+HIV_-induced LX2 activation [45]. As a result, STAT3 was inhibited in LX2 cells by STAT3 siRNA transfection to evaluate the direct influence of STAT3 on AB_AGS+HIV_-induced LX2 activation. We found the attenuation of profibrotic genes: ACTA2 (Figure 10A), COL1A1 (Figure 10B), and TGFβ1 (Figure 10D). Furthermore, MMP2 mRNA, which encodes for the synthesis of a matrix-degrading enzyme, was upregulated by 50% (Figure 10C). These findings were paralleled by immunoblotting analysis of α-SMA in STAT3-silenced LX2 cells exposed to AB_AGS+HIV_. α-SMA in these LX2 cells was reduced by approximately 43% when compared to not transfected LX2 cells (Figure 10E,F).

### 3.10. In Vivo Effects of HIV Containing-Apoptotic Bodies on Ethanol-Fed Mice

To validate the in vitro effects of HIV-derived ABs in vivo, we used immunodeficient NSG mice. The NSG mice were fed ethanol or control diet for 10 weeks and intravenously injected with HIV-containing human hepatocyte (RLW) ABs (0.5–1.0 × 10^7^ ABs /mouse) during the last two weeks of ethanol feeding. To induce liver fibrosis, the mice were injected with CCl4 (i.p). At week 10, mice were euthanized. As shown in Figure 11A,B, ethanol-fed mice injected with HIV-containing RLW ABs showed elevation of profibrotic genes (TGFβ2 and COL1A1) in liver tissue. Further analysis revealed elevation of liver enzymes, AST and ALT, in the same mice (Figure 11C,D). Deposition of ECM component (collagen) was also observed in ethanol-fed mice injected with HIV-containing ABs (Figure 11E). 

## 4. Discussion

Mortality due to end-stage liver disease is on the rise among people living with HIV [46]. Clinical studies suggest that alcohol, antiretroviral therapy and hepatotropic viruses (Hepatitis B and C) may be associated with liver disease among HIV-infected individuals [47]. While the mechanisms of hepatotoxicity by antiretroviral therapy [48,49] and hepatotropic viruses [50,51] have been extensively explored, the mechanisms explaining the role of alcohol in HIV-infection and pathogenesis of liver disease development has not received adequate attention despite the emerging burden of liver disease in HIV management [52]. In response, this study explored preclinical experimental designs to decipher ethanol-induced mechanisms in hepatocytes exposed to HIV. 

The motivation for addressing the specific aims of this study emanated from the profound relationship between hepatocyte apoptosis and liver fibrosis [53]. In fact, previous studies have reversed the progression of liver fibrosis by blocking hepatocyte apoptosis [54,55,56,57]. In our hands, profibrotic activation of HSC became even more prominent when hepatocytes, as a source of ABs, expressed HIV and oxidative stress markers. While numerous factors have been identified as triggers for hepatocyte apoptosis, the combination of ethanol and HIV was clearly demonstrated by us as a potent trigger of hepatocyte apoptosis via oxidative stress. This is not surprising because hepatocytes predominantly metabolize ethanol by ADH to generate the most toxic oxidative metabolite, acetaldehyde [58]. Acetaldehyde is the major trigger for HIV accumulation and the consequent hepatocyte apoptosis as reported in our previous study [13]. However, ADH is not the only ethanol-metabolizing enzyme potentiating HIV-induced hepatocyte apoptosis. In addition, the microsomal ethanol oxidizing system (MEOS) consisting of CYP2E1 may also be involved [59,60]. In fact, we previously showed in primary human hepatocytes that the combined treatment of ethanol and HIV which engendered the highest apoptosis upregulates CYP2E1 expression as well [13]. Here, as a source of hepatocytes, we used RLW cells, which are Huh7.5 cells stably transfected with CYP2E1. These cells do not express ADH and, thus, do not make a sufficient amount of acetaldehyde. To mimic ethanol metabolism in primary human hepatocytes, we exposed RLW cells to an acetaldehyde-generating system (AGS). Since ethanol is a part of this system, exposure of CYP2E1-overexpressing cells to ethanol allows stabilizing CYP2E1 under ethanol treatment and generates ROS [61]. This in a combination with acetaldehyde continuously produced by AGS provides hepatotoxic effects leading to hepatocyte AB formation. Our previous studies on HIV-infected hepatocytes exposed to ethanol demonstrated that these hepatotoxic effects can be reversed by 4-methyl pyrazole, an ADH inhibitor [13]. Here, we also demonstrated that antioxidant NAC reverses profibrotic activation of HSC after engulfment of HIV+AGS-induced AB, indicating the role of CYP2E1-generated ROS in liver fibrosis development [62]. 

Furthermore, our recent studies uncovered how the combined treatment with acetaldehyde, an ethanol metabolite, and HIV triggered hepatocyte accumulation of p24 and HIV gag RNA, which resulted in oxidative stress-induced apoptosis of HIV-infected hepatocytes via lysosome impairment [13,14]. Although the apoptotic death of HIV-infected hepatocytes may be considered beneficial for the reduction of the HIV liver reservoir, it was concurrently attributed to profibrotic changes in the liver. Large EVs, particularly ABs derived from apoptotic hepatocytes, were found to mediate the observed profibrotic changes, which are specific to ABs of hepatocyte origin, while lymphocyte-derived ABs elicited no profibrotic changes in HSC. To substantiate that the observed profibrotic effects of ABs on HSC were attributed to HSC and are not directly from the internalized ABs, AB-induced profibrotic genes in HSC were measured in the hepatocyte AB and they could not be determined. Therefore, the observed profibrotic genes were connected to HSC. 

Although ABs in other studies were generated by different apoptotic stimuli, such as viral agents (hepatitis C virus (HCV) [50,63], HIV [13]), ultraviolet light [64], and toxic substances, namely CCl4 [65] and ethanol [66], in the current study ABs were generated by the combined treatment of acetaldehyde and HIV, hence closely mimicking the situation in alcohol-abusing PLWH. This provides an appropriate model to study frequently observed liver fibrosis in alcohol-abusing HIV-infected individuals. Therefore, aiming to explore the mechanisms of liver fibrosis in the context of HIV and alcohol, we hypothesize that hepatocyte exposure to the combination of HIV and acetaldehyde ends with the generation of HIV-containing ABs that activate oxidative stress triggered profibrotic changes in HSC. 

HSCs are specialized cells that generate ECM to replace dead hepatocytes. While this process is a well-defined homeostatic process, it becomes pathological when hepatocyte death is excessive. HSC activation to myofibroblast has been demonstrated by HSC engulfment of ABs in this and other studies [67,68]. Although it may be argued that macrophages are better professional “eaters” than HSC [69], the anatomical proximity of HSC to hepatocytes in the liver architecture is favorable for HSCs to encounter hepatocyte ABs. Moreover, the combination of HIV and acetaldehyde induces massive hepatocyte apoptosis [13], allowing non-professional (HSC) phagocytosis of hepatocyte AB_AGS+HIV_. So, it is logical to design an experiment involving the exposure of HSC to ABs to study AB-induced HSC activation. While AB engulfment capacity and the consequential profibrotic activation in HSC is well studied, the AB cargo-dependent mechanism of HSC activation in HIV infection needs to be clarified. Thus, this study focused on characterizing AB_AGS+HIV_, with a special focus on measuring AB contents. Here, we observed significant expression of HIV gag RNA, HIV proteins and MDA both in ABs and in HSC after AB internalization, suggesting an important role of ABs in the crosstalk between liver parenchymal and non-parenchymal cells. Although the HIV product transfer did not elicit productive HIV infection in HSC or even permissive immune cells, both the HIV products and the oxidative stress markers (MDA, ROS) promoted HSC activation as indicated by the upregulation of profibrotic genes in LX2 cells exposed to ABs obtained from both acetaldehyde- and HIV-exposed RLW cells. 

Externalized PtylS, robustly expressed by ABs, is known as the “eat me” signal for HSC engulfment of ABs [70]. While HSC internalization of ABs was previously demonstrated using an array of scavenger receptors of both A and B classes [71], our data confirmed that HSC engulfment of ABs is possible through one of the receptor tyrosine kinases (RTK). In fact, HSCs express Gas 6 and Protein S, which are required as an insert to complete ABs internalization and the engulfment of TAMRA-labeled ABs, followed by profibrotic gene activation in LX2 cells, was significantly attenuated by Axl inhibitor. Given that Axl and Gas6 are notoriously involved in ABs engulfment by liver resident macrophages [72,73,74], our observation of Axl, Gas 6 and Protein S involvement in HSC engulfment of AB_AGS+HIV_ confirms that this mechanism works for HSCs exposed to ABs. 

While many studies agree that HSC engulfment of ABs is an essential trigger of profibrotic activation, the next question is which signaling pathways are responsible for this HSC activation after AB_AGS+HIV_ engulfment. Previous studies identified the involvement of p38 MAPK in the activation of HSC [71]. However, our data on the engulfment of AB_AGS+HIV_ conversely observed profibrotic activation through the ERK1/2 pathway after the inhibition of p38 MAPK. In addition to p38 MAPK, we also tested the role of JNK, another MAPK upstream from ERK1/2. JNK inhibition attenuated profibrotic activation as well as the secretion of ECM component α-SMA in LX2 cells. This finding is in congruence with previous studies that reported the role of JNK in hepatic fibrogenesis [75,76], even though ABs did not induce JNK activation in those studies. While it may not be clear why the MDA and HIV-containing AB_AGS+HIV_ activated JNK, we are inclined to lean towards MDA as being partly involved in JNK activation. In fact, a previous study identified 4-HNE-protein-adducts, another oxidative product, as a stimulus for JNK activation in HSC [77].

Moreover, our data revealed that the engulfment of HIV and MDA-containing AB_AGS+HIV_ led to ROS generation in HSC. In addition, HSC activation via the JNK-ERK1/2 pathways after AB_AGS+HIV_ engulfment was attenuated by NAC, a known antioxidant. This is counterintuitive because ROS are pro-apoptotic in cells, while, here, ROS are paradoxically anti-apoptotic since they promote cell survival, proliferation, and profibrotic activation in HSC [78,79,80,81,82]. 

As expected, HSC activation after AB_AGS+HIV_ engulfment involved multiple pathways. JAK-STAT3-mediated HSC activation after ABs engulfment was observed through ROS generation [67]. Similarly, our data demonstrated HSC activation via the JAK-STAT3 pathway after AB_AGS+HIV_ engulfment. In fact, STAT3 gene inhibition significantly attenuated α-SMA and profibrotic genes in HSCs. Moreover, the upregulation of IL6 mRNA in HSC after AB_AGS+HIV_ engulfment and the expression of IL6 receptors on HSC [83] confirms that AB_AGS+HIV_-induced JAK-STAT3 activation in HSC may be induced by IL6, a canonic ligand for STAT3 activation [84]. In addition, SOCS1, a suppressor of STAT3 activation [85], was downregulated by AB_AGS+HIV_ internalization, which increases JAK-STAT3 signaling in LX2 cells; this signaling is pro-survival for HSC and is known to interfere with anti-viral IFN type 1 signaling, thereby attenuating activation of anti-viral ISGs.

A major limitation to our experimental design was lack of primary human hepatocytes, which were not available in necessary quantities to generate the substantial amount of hepatocyte ABs required for this study. However, HIV and ethanol-induced hepatocyte apoptosis observed in RLW cells and its downstream effects on LX2 cells were previously induced by ethanol in primary human hepatocytes [13]. In this study, LX2 cells were grown on plastic plates, a known trigger for LX2 activation; however, this did not affect AB-induced HSC activation because the AB-unexposed LX2 cells were also cultured on plastic plates as LX2 cells exposed to AB. 

AB-induced HSC activation within the context of ethanol metabolites and HIV was validated in vivo by injecting ethanol-fed immunodeficient NSG mice with HIV-containing ABs. In this experiment, we observed the upregulation of TGFβ2 and COL1A1 genes which led to the deposition of collagen in the liver of ethanol fed mice injected with hepatocyte AB_HIV_, and this was accompanied by elevated liver enzymes. Our major findings are summarized in Figure 12.

## 5. Conclusions

In summary, we observed a massive generation of HIV and MDA-containing ABs when hepatocytes were treated with the combination of acetaldehyde and HIV. When these ABs were exposed to HSCs, no productive HIV infection was elicited; however, massive ROS release was generated, leading to HSC activation through the JNK-ERK1/2 and JAK-STAT3 pathways. We believe that these results provide a mechanistic insight on alcohol- and HIV-induced liver fibrosis development. Therefore, a therapeutic regimen with antioxidant properties should be effective in ameliorating the progression of liver fibrosis among alcohol-abusing PLWH. 

## Figures and Tables

**Figure 1 biology-11-01059-f001:**
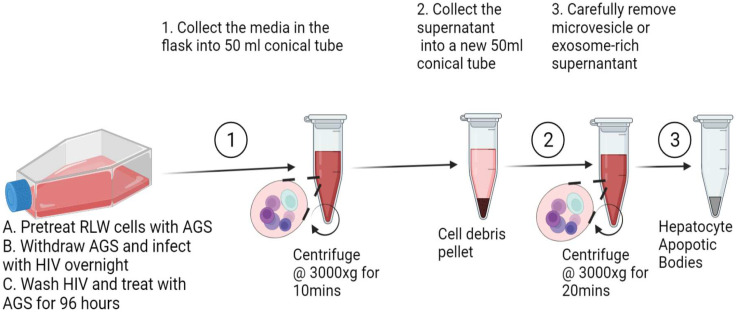
Schematic drawing of the generation and isolation of ABs derived from acetaldehyde and HIV-exposed RLW cells using a differential centrifugation method.

**Figure 2 biology-11-01059-f002:**
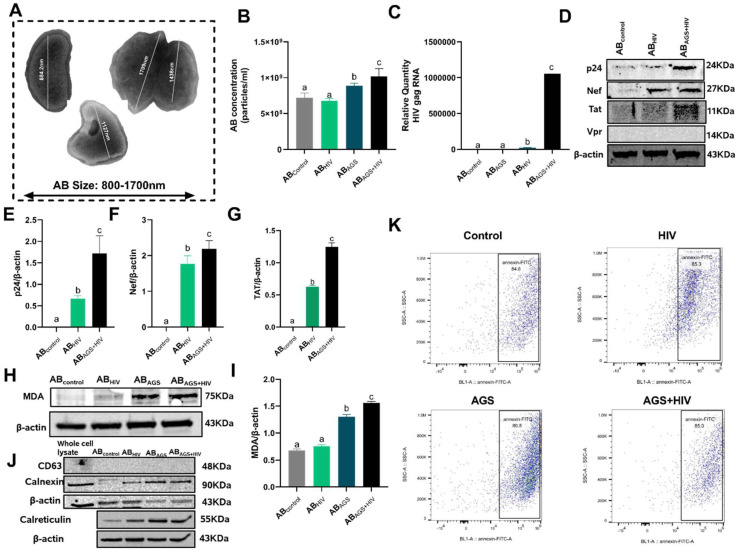
Characterization of ABs derived from acetaldehyde and HIV-exposed RLW: (**A**) Sizes of RLW ABs visualized by transmission electron microscopy. (**B**) Quantification of RLW ABs by NTA. (**C**) HIV gag RNA expression in RLW ABs quantified by RT-PCR analysis. (**D**) HIV proteins: p24, Nef, and Tat in RLW ABs were measured by immunoblotting analysis. Equal amounts of protein (20 µg) were loaded in each lane. β-actin was used as an internal control. (**E**–**G**) Quantification of the immunoreactive bands of HIV proteins: p24, Nef, and Tat in RLW Abs. (**H**,**I**) Malondialdehyde (MDA) content in RLW ABs was measured by immunoblotting analysis and quantification of immunoreactive protein bands was provided. Equal amounts of protein (20 µg) were loaded in each lane. β-actin was used as an internal control. (**J**) Exosome marker (CD63) and AB markers (calnexin and calreticulin) were measured by immunoblotting analysis of RLW AB lysate. Equal amounts of protein (20 µg) were loaded in each lane. (**K**) Quantification of phosphatidylserine surface expression on RLW ABs measured by flow cytometry. Western blot detected bands were from the representative experiments. Statistically processed data are from 3 independent experiments presented as mean ± SEM. Bars marked with the same letter are not significantly different; bars with different letters are significantly different from each other (*p* ≤ 0.05).

**Figure 3 biology-11-01059-f003:**
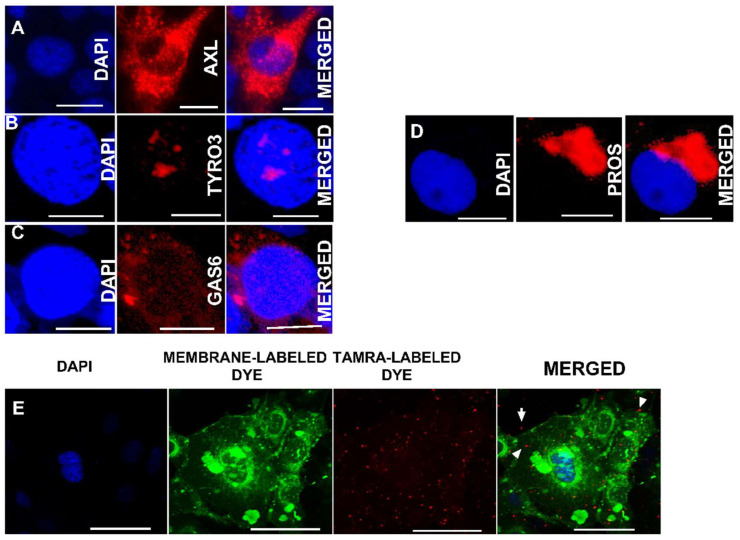
Gas6 and Protein S mediated engulfment of RLW ABs by the AXL-expressing LX2 cells: Immunofluorescent staining of (**A**) Axl receptors, (**B**) Tyro3 receptors, (**C**) Gas 6, (**D**) Protein S, (**E**) Immunofluorescence staining to demonstrate the engulfment of RLW ABs by LX2 cells. RLW ABs were labeled with TAMRA dye and LX2 membranes were stained with cytoplasmic dye (CellBrite^®^ Cytoplasmic Membrane Dyes, San Francisco, USA). Staining was visualized using 20× lens of a Keyence BZ-X810 fluorescence microscope. Pictures are of representative data from three independent experiments.

**Figure 4 biology-11-01059-f004:**
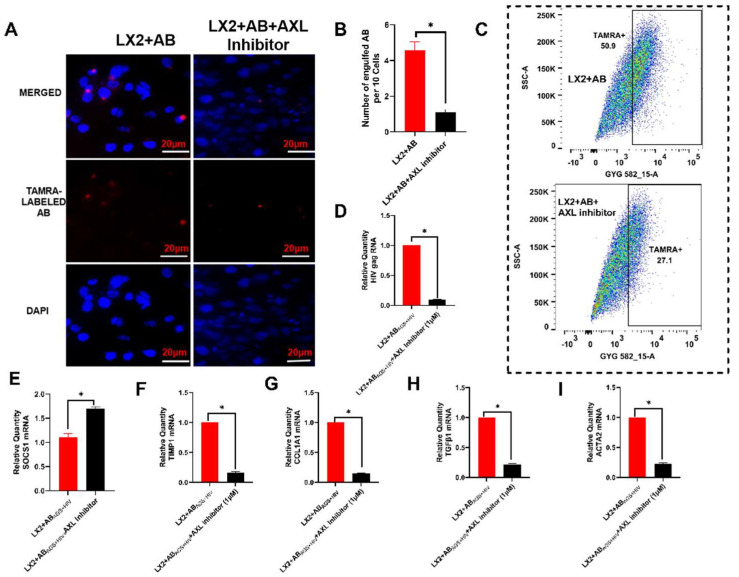
AXL inhibitor attenuates HIV RNA expression and profibrotic genes in LX2 cells exposed to AB_AGS+HIV_: (**A**) Immunofluorescence staining demonstrates the attenuation of TAMRA-labeled AB entry into LX2 cells pretreated with Axl-inhibitor. Staining was visualized using a 20× BZ-X810 fluorescence microscope. Pictures are the representative data of three independent experiments. (**B**) Quantification of engulfed TAMRA-labelled RLW ABs by LX2 cells. Engulfment of TAMRA-labelled RLW ABs by LX2 cells was counted manually in 3 fields. (**C**) Intracellular quantification of TAMRA-labelled RLW ABs exposed to LX2 cells (flow cytometry). RT PCR analysis of: (**D**) HIV gag RNA, (**E**) SOCS1 mRNA, (**F**) TIMP1 mRNA, (**G**) COL1A1 mRNA, (**H**) TGFβ1 mRNA, and (**I**) ACTA2 mRNA. Data are from 3 independent experiments presented as mean ± SEM. * Indicates significant difference (*p* ≤ 0.05).

**Figure 5 biology-11-01059-f005:**
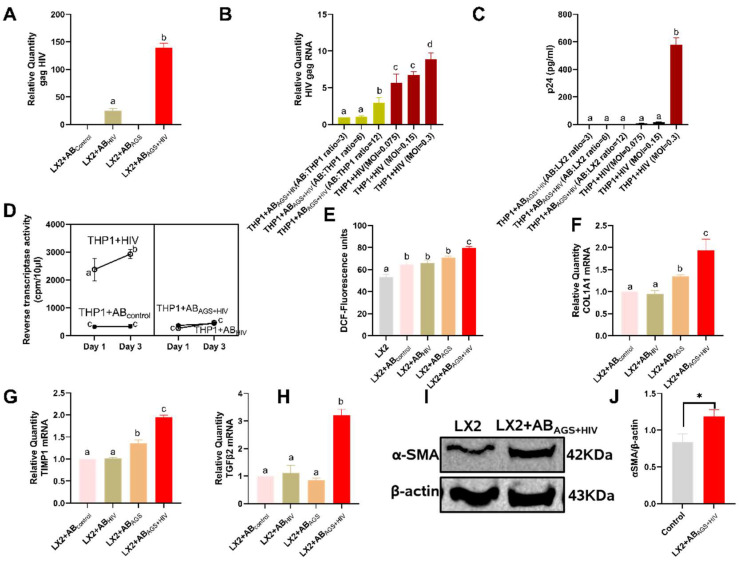
Engulfment of AB_AGS+HIV_ enhances HIV RNA and ROS in LX2 cells leading to their profibrotic activation: (**A**) RT-PCR analysis of HIV gag RNA expression in LX2 cells after engulfment of AB_AGS+HIV_. (**B**) RT-PCR analysis of HIV gag RNA expression in THP-1 cells after 48 h exposure to RLW AB_AGS+HIV_. (**C**) Quantification of HIV p24 levels in the culture media from THP1 exposed to RLW AB_AGS+HIV_ for 48 h by ELISA. (**D**) RT activity in THP-1 cells exposed to RLW AB_AGS+HIV_. Open dots represent THP1 cells exposed to HIV1_ADA_; closed dots represent THP1 cells exposed to Abs. (**E**) ROS generation after 48 h exposure of LX2 cells to RLW AB_AGS+HIV_ (2′,7′-dichlorodihydrofluorescein diacetate assay). RT-PCR analysis of (**F**) COL1A1 mRNA, (**G**) TIMP1 mRNA, and (**H**) TGFβ2 mRNA of LX2 cells exposed to RLW AB_AGS+HIV_ for 2 h. (**I**,**J**) α-Smooth muscle actin (SMA) of LX2 cells exposed to RLW AB_AGS+HIV_ for 48 h was measured by immunoblot analysis and the quantification of immunoreactive protein bands. Equal amounts of protein (20 µg) were loaded in each lane. β-actin was used as an internal control. Data are from 3 independent experiments presented as mean ± SEM. Bars marked with the same letter are not significantly different; bars with different letters are significantly different from each other. Bars with * are significantly different from each other (*p* ≤ 0.05).

**Figure 6 biology-11-01059-f006:**
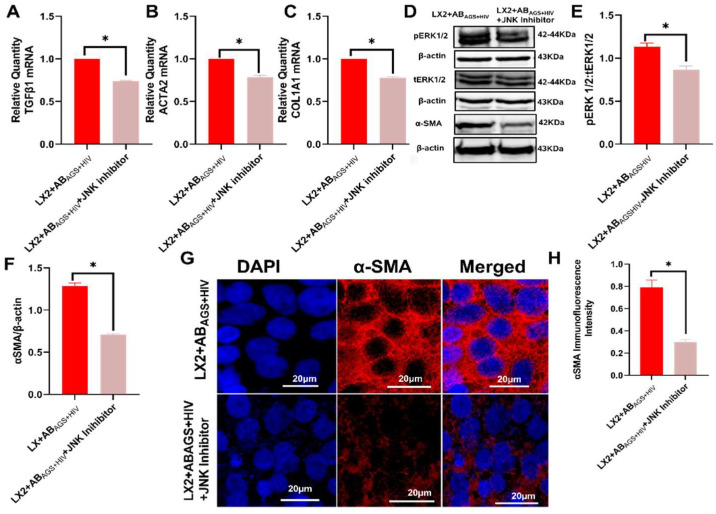
Attenuation of profibrotic markers in LX2 cells exposed to AB_AGS+HIV_ by pharmacological inhibition of JNK pathway: RT-PCR analysis of (**A**) TGFβ1 mRNA, (**B**) ACTA2 mRNA, and (**C**) COL1A1 mRNA. (**D**–**F**) Phopho-ERK1/2 and α-smooth muscle actin were measured by immunoblot analysis and immunoreactive blots were quantified. Equal amounts of protein (20 µg) were loaded in each lane. β-actin was used as an internal control. (**G**) Immunofluorescent staining of α-smooth muscle actin. Staining was visualized using a Keyence BZ-X810 fluorescence microscope. Pictures are representative data from three independent experiments. (**H**) Quantification of SMA was measured using NIH ImageJ. Data are from 3 independent experiments presented as mean ± SEM. Bars marked with the same letter are not significantly different. Bars with * are significantly different from each other (*p* ≤ 0.05).

**Figure 7 biology-11-01059-f007:**
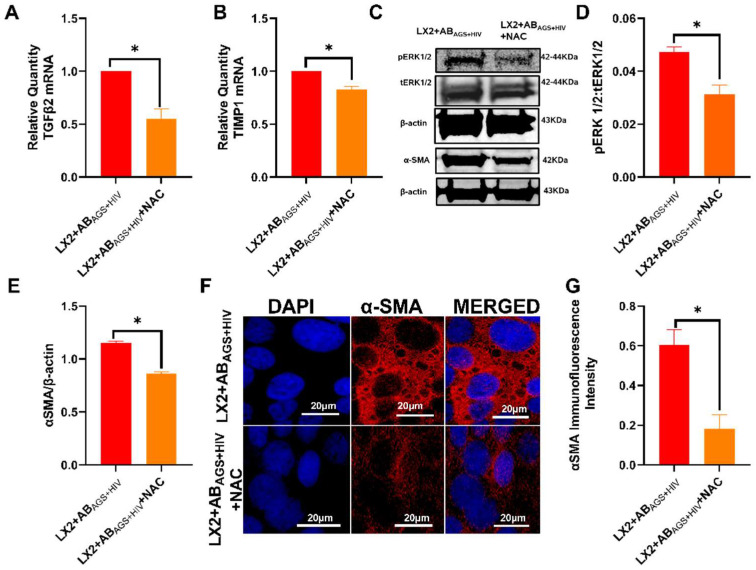
NAC treatment reverses or prevents profibrotic activation of LX2 cells by inhibition of ERK1/2 pathway: RT-PCR analysis of (**A**) TGFβ2 mRNA and (**B**) TIMP1 mRNA. (**C**–**E**) Phopho-ERK1/2 and α-SMA were measured by immunoblot analysis and the quantification of immunoreactive blots. Equal amounts of protein (20 µg) were loaded in each lane. β-actin was used as an internal control. (**F**) Immunofluorescent staining of α-SMA. Staining was visualized using a Keyence BZ-X810 fluorescence microscope. Pictures are representative data from three independent experiments. (**G**) Quantification of α-SMA by NIH ImageJ. Data are from 3 independent experiments presented as mean ± SEM. Bars with * are significantly different from each other (*p* ≤ 0.05).

**Figure 8 biology-11-01059-f008:**
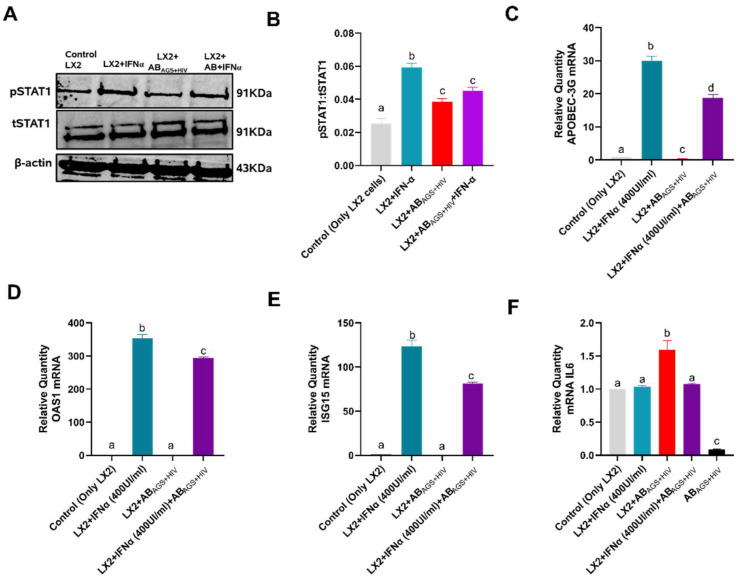
Engulfment of AB_AGS+HIV_ attenuates STAT1 activation and concurrently upregulates IL6 mRNA, leading to profibrotic activation: (**A**,**B**) STAT1 was measured by immunoblot analysis and the quantification of immunoreactive protein bands. Equal amounts of protein (20 µg) were loaded in each lane. β-actin was used as an internal control. RT-PCR analysis of (**C**) APOBEC-3G mRNA, (**D**) OAS1 mRNA, (**E**) ISG15 mRNA, and (**F**) IL6 mRNA. Data are from 3 independent experiments presented as mean ± SEM. Bars marked with the same letter are not significantly different. Bars with different letters are significantly different from each other (*p* ≤ 0.05).

**Figure 9 biology-11-01059-f009:**
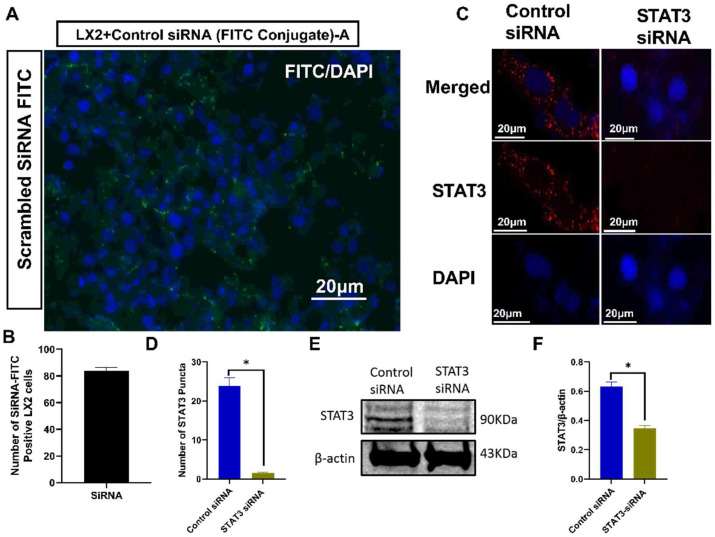
SiRNA transfection of LX2 cells: (**A**) Transfection of LX2 cells with scrambled siRNA-FITC (green). Visualized using a 20× Keyence microscope. Pictures are representative data from three independent experiments. (**B**) Quantification of transfection efficiency. (**C**) Immunofluorescent staining of STAT3. Staining was visualized using a 20× lens in Keyence BZ-X810 florescence microscope. Pictures are representative data from three independent experiments. (**D**) STAT3 immunofluorescence intensity was measured using NIH ImageJ. (**E**,**F**) STAT3 was measured by immunoblot analysis and the quantification of immunoreactive protein bands. Equal amounts of protein (20 µg) were loaded in each lane. β-actin was used as an internal control. Data are from 3 independent experiments presented as mean ± SEM. Bars with * are significantly different from each other (*p* ≤ 0.05).

**Figure 10 biology-11-01059-f010:**
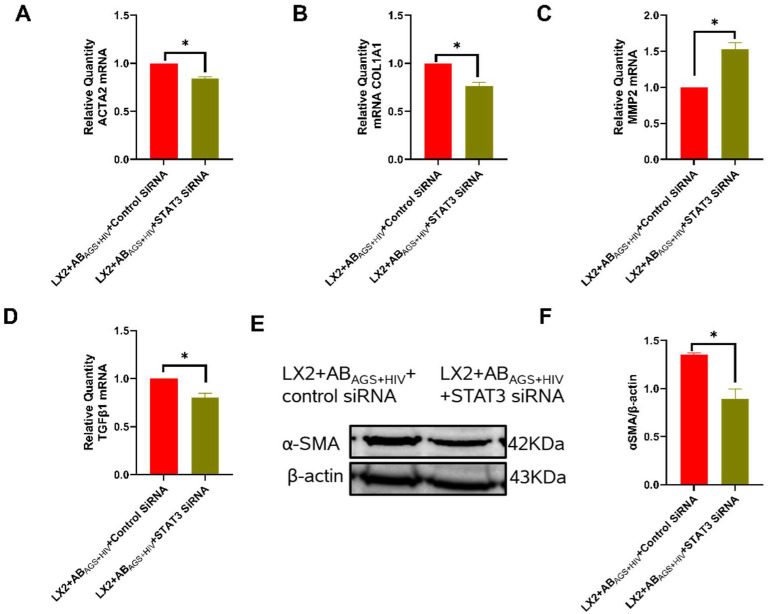
STAT3-deficient LX2 cells show downregulation of profibrotic genes and α-SMA after exposure to AB_AGS+HIV_: RT-PCR analysis of: (**A**) ACTA2 mRNA, (**B**) COL1A1 mRNA, (**C**) MMP2 mRNA, and (**D**) TGFβ1 mRNA. (**E**,**F**) α-SMA was measured by immunoblot analysis and the quantification of immunoreactive protein bands. Equal amounts of protein (20 µg) were loaded in each lane. β-actin was used as an internal control. Data are from 3 independent experiments presented as mean ± SEM. Bars with * are significantly different from each other (*p* 0.05).

**Figure 11 biology-11-01059-f011:**
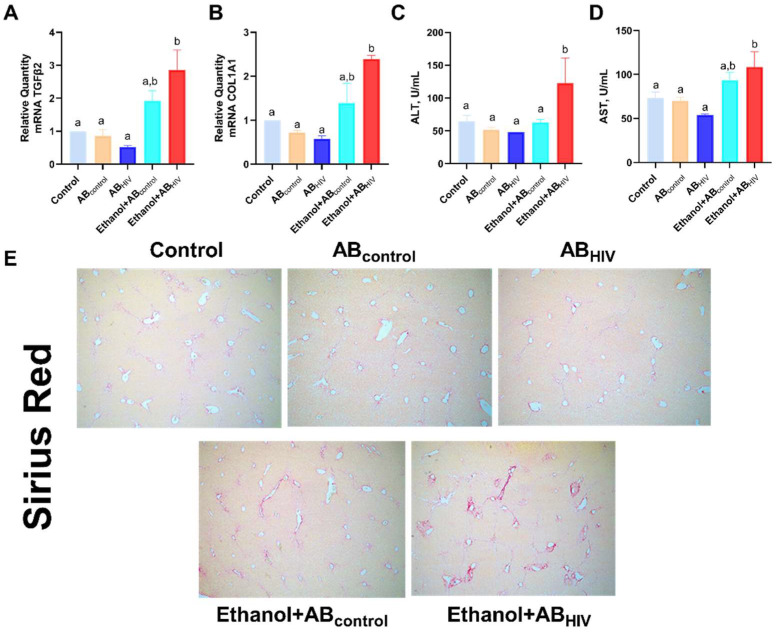
Ethanol potentiates collagen deposition and liver damage in the liver of mice treated with HIV-containing apoptotic bodies: RT-PCR analysis of mouse tissues for (**A**) TGFβ2 mRNA and (**B**) COL1A1 mRNA. (**C**) ALT levels (U/mL); (**D**) AST levels (U/mL). (**E**) Histological examination of mouse liver tissues using Sirius red staining. Bars marked with the same letter are not significantly different. Bars with different letters are sig-nificantly different from each other (*p* ≤ 0.05).

**Figure 12 biology-11-01059-f012:**
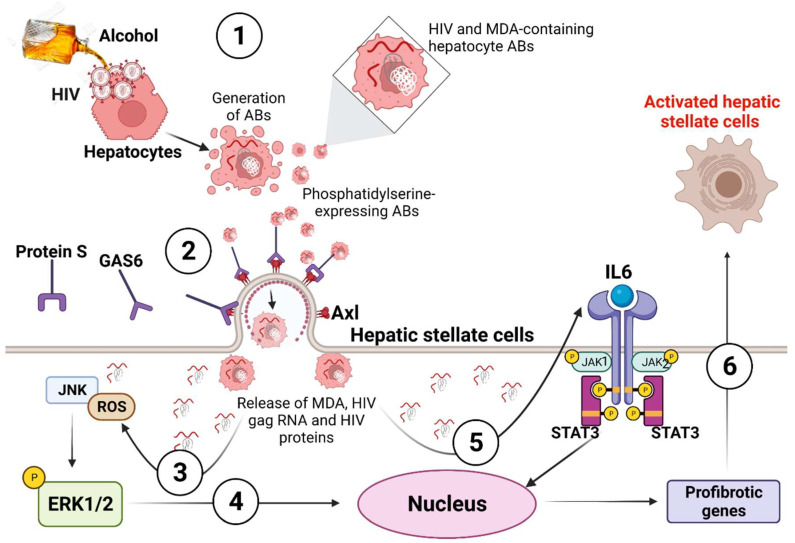
HSC internalization of ABs generated from alcohol- and HIV-exposed hepatocytes induced ROS and IL6-mediated HSC activation. The mechanistic steps include: 1. Both the alcohol metabolite, acetaldehyde, and HIV trigger hepatocyte apoptosis. This leads to the generation of HIV and MDA-containing hepatocyte ABs which express PtylS. 2. HSC internalizes HIV and MDA-containing ABs through Axl, a PtylSRR, with the help of the two bridging molecules, Gas 6 and protein S. 3. The release of HIV proteins, HIV gag RNA and MDA into HSC by internalized ABs triggers intracellular ROS generation. This ROS mediates JNK activation, which leads to the phosphorylation of the ERK1/2 pathway. 4. Nuclear translocation of phosphorylated ERK1/2 induced the biosynthesis of profibrotic genes. 5. The delivery of HIV and MDA into HSC by internalized ABs induced IL-6 synthesis required to activate the JAK-STAT3 pathway. The activated STAT3 leads to the biosynthesis of profibrotic genes. 6. Activation of profibrotic genes in HSC promotes liver fibrosis development.

## Data Availability

Not applicable.

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
