# Peer review of "Alcohol and HIV-Derived Hepatocyte Apoptotic Bodies Induce Hepatic Stellate Cell Activation"

_biology, 2022, doi:10.3390/biology11071059_

Round 1

Reviewer 1 Report

This is an interesting study of alcohol and HIV on activation of HSC  alon without considering other esucellular areas of the hepatocyte. Substantial arguments  should be considered.

Major points:

1. How sure are you that the reported results are connected with HSC? Please discuss in the text.

2. Please add a para in the discussion section on alcohol metabolism via hepatic MEOS and CYP 2E1 within the liver endoplasmic reticulum, conditions that also produce ROS.

3. Add more clinical aspects on ALD in the text.

Author Response

Dear Reviewers,

Thank you for the timely and rigorous review of our manuscript. We have made changes to our manuscript based on your suggestions. Please find in the paper the changes made in red fonts. 

Major points:

  1. How sure are you that the reported results are connected with HSC? Please discuss this in the text.

Response: The reported results showed the upregulation of profibrotic genes in HSC after the internalization of AB (apoptotic bodies). To ensure that the upregulated genes are connected to HSC, we quantified the profibrotic genes in the hepatocyte ABs (Supplementary D3&4). The profibrotic genes expressed by HSC upon internalization of AB were undetermined in hepatocyte AB. Furthermore, HSC activation after internalization of hepatocyte AB is well reported (PMID: 16496318; PMID: 16496318; PMID: 16258409), but not in the context of alcohol and HIV, which provide more robust changes than control (HIV-AGS non-exposed) AB As recommended, this observation has been added to the discussion section of the text (page 10, lines 389-390 and page 17, lines 596-600)

  1. Please add a para in the discussion section on alcohol metabolism via hepatic MEOS and CYP 2E1 within the liver endoplasmic reticulum, conditions that also produce ROS.

Response: The authors agree with the reviewer that hepatic MEOS and CYP2E1 may be involved in ROS generation of ethanol-exposed HIV-containing hepatocytes. This may be contributing to hepatocyte apoptosis and generation of AB. An update concerning this comment was added to page 6 line 296 & page 16-17 lines 567-587. The role of these CYP2E1-induced ROS in activation of profibrotic genes in HSC is confirmed by reversive effects of ROS scavenger, NAC.

  1. Add more clinical aspects on ALD in the text.

Response: The authors have added the clinical significance of ALD among people living with HIV to the discussion (page 16 lines 551-559).

Reviewer 2 Report

In the manuscript entitled “Alcohol and HIV-derived Hepatocyte Apoptotic Bodies Induce  Hepatic Stellate Cell Activation ”, the authors reported the importance of alcohol and HIV-derived hepatocyte apoptotic bodies on liver fibrosis.
While some of the observations are interesting, this work is preliminary and the manuscript remains descriptive.

There are some concerns about the experiments:

1. Apoptotic bodies were collected, but were their particle size measured? Was there any contamination such as exosomes?

2. Fig. 3 shows the incorporation of Apoptotic Bodies into HSCs, what was the efficiency of the incorporation?

3. Has LX2 been cultured on plastic plates? Was it already in an activated state? Was the response to apoptotic bodies the same for HSCs in the quiescent and activated state?

4. In vivo studies, did the authors confirm that the administered apoptotic bodies were taken up by hepatic stellate cell? Besides the hepatic stellate cells, was there any incorporation to Kupffer cells or other tissues outside of the liver? What percentage of the apoptotic bodies administered was taken up by HSCs?

5. In the simple summary, it says that therefore, a potential therapeutic regimen for reducing liver fibrosis among PLWH should include ROS-scavenging antioxidants. but where is the data showing that Scavenger is effective? What kind of scavenger is preferable?

Author Response

Dear Reviewer,

Thank you for the timely and rigorous review of our manuscript. We have made changes to our manuscript based on your suggestions. Please find the changes on the manuscript in red fonts:

  1. Apoptotic bodies were collected, but were their particle size measured? Was there any contamination, such as exosomes?

Response: The particle size of the apoptotic bodies was measured and reported in Figure 2A. The size ranges between 800nm and 1700nm, which is typical for apoptotic bodies—the method of isolation by differential centrifugation excluded possible exosome contamination. In addition, isolated apoptotic bodies are intensively washed in a serum-free medium, to avoid contamination. Moreover, to substantiate the purity of the apoptotic bodies, we probed for the exosome marker anti-CD63 via western blotting and found no CD63 band, while the markers of apoptotic proteins (calnexin, calreticulin and phosphatidylserine) were expressed

  1. Fig. 3 shows the incorporation of Apoptotic Bodies into HSCs; what was the efficiency of the incorporation?

Response: A meticulous visualization and quantification of the internalized AB in Fig 3 revealed more than 90% AB internalization efficiency by HSCs. Moreover, we are not the first to provide evidence for the HSC internalization of AB. Previous studies have demonstrated this (doi.org/10.1097/01.LAB.0000069036.63405.5C). 

  1. Has LX2 been cultured on plastic plates? Was it already in an activated state? Was the response to apoptotic bodies the same for HSCs in the quiescent and activated state?

Response: LX2 cells were cultured on plastic plates, and we do not ignore that plastic plates may activate the HSCs. In response to this, we have included in the text that LX2 cells without AB were partially activated, as indicated in Figure 5I. However, AB derived from AGS-and HIV-exposed hepatocytes increased the observed profibrotic changes. (We addressed this on page 21 lines 675-678)

  1. In vivo studies, did the authors confirm that the administered apoptotic bodies were taken up by hepatic stellate cells? Besides the hepatic stellate cells, was there any incorporation of Kupffer cells or other tissues outside the liver? What percentage of the apoptotic bodies administered was taken up by HSCs?

Response: Given that the NSG mice used for this study were immunodeficient, they lack lymphocytes (NK cell-deficient). While the number of Kupffer cells was unchanged, their functions were impaired (PMID: 7995938), so we do not expect active engulfment of ABs by Kupffer cells. While the scope of this study did not track the number of apoptotic bodies taken up by HSCs, remarkable profibrotic damage was observed when the mouse livers was exposed to AB. We observed an activation of fibrotic changes after injection of AB, and as known, these profibrotic changes are associated with HSC activation.

  1. In the simple summary, it says that therefore, a potential therapeutic regimen for reducing liver fibrosis among PLWH should include ROS-scavenging antioxidants. but where is the data showing that Scavenger is effective? What kind of scavenger is preferable?

Response: This conclusion was made because we found correlations between AB-induced ROS generation and AB-induced profibrotic changes. Moreover, treatment of LX2 cells exposed to AB with N-acetyl cysteine (Figure 7), a glutathione precursor and an antioxidant, attenuated the observed profibrotic changes.

Round 2

Reviewer 1 Report

Thank you for revision.  Text is now ok, however, please include appropriate actual references for: MEOS, ROS, and CYP2E1.

Author Response

Dear Reviewer,  

Thank you for promptly returning our manuscript for the second round of revision. We agree that additional references are needed to support our discussion on MEOS, CYP2E1, and ROS. We added the following references per your recommendations.    

58.          Guo, R.; Ren, J., Alcohol and acetaldehyde in public health: from marvel to menace. Int J Environ Res Public Health 2010, 7 (4), 1285-301. 

59.          Teschke, R.;  Neuman, M. G.;  Liangpunsakul, S.; Seitz, H.-K., Alcoholic Liver Disease and the co-triggering Role of MEOS with Its CYP 2E1 Catalytic Cycle and ROS. Archives of Gastroenterology Research 2021, 2 (1), 9-25. 

60.          Doody, E. E.;  Groebner, J. L.;  Walker, J. R.;  Frizol, B. M.;  Tuma, D. J.;  Fernandez, D. J.; Tuma, P. L., Ethanol metabolism by alcohol dehydrogenase or cytochrome P(450) 2E1 differentially impairs hepatic protein trafficking and growth hormone signaling. Am J Physiol Gastrointest Liver Physiol 2017,313 (6), G558-g569. 

61.          Koop, D. R., Alcohol metabolism's damaging effects on the cell: a focus on reactive oxygen generation by the enzyme cytochrome P450 2E1. Alcohol Res Health 2006, 29 (4), 274-80. 

62.          Urtasun, R.;  de la Rosa, L. C.; Nieto, N., Oxidative and nitrosative stress and fibrogenic response. Clinics in liver disease 2008, 12 (4), 769-790. 

Reviewer 2 Report

Thank you for submitting a revised version of the manuscript.

All my concerns were cleared.

Author Response

Thank you for the acknowledgment!

Round 3

Reviewer 1 Report

Thank you for revision.